# Prognostic Significance of Interferon-γ and Its Signaling Pathway in Early Breast Cancer Depends on the Molecular Subtypes

**DOI:** 10.3390/ijms21197178

**Published:** 2020-09-29

**Authors:** Anne-Sophie Heimes, Franziska Härtner, Katrin Almstedt, Slavomir Krajnak, Antje Lebrecht, Marco J. Battista, Karolina Edlund, Walburgis Brenner, Annette Hasenburg, Ugur Sahin, Mathias Gehrmann, Jan G. Hengstler, Marcus Schmidt

**Affiliations:** 1Department of Obstetrics and Gynecology, University Medical Center Mainz, 55131 Mainz, Germany; anne-sophie.heimes@unimedizin-mainz.de (A.-S.H.); katrin.almstedt@unimedizin-mainz.de (K.A.); slavomir.krajnak@unimedizin-mainz.de (S.K.); antje.lebrecht@unimedizin-mainz.de (A.L.); marco.battista@unimedizin-mainz.de (M.J.B.); walburgis.brenner@unimedizin-mainz.de (W.B.); annette.hasenburg@unimedizin-mainz.de (A.H.); 2Institute of Medical Biostatistics, Epidemiology and Informatics (IMBEI), University Medical Center Mainz, 55131 Mainz, Germany; f.haertner@uni-mainz.de; 3Leibniz-Research Centre for Working Environment and Human Factors at the TU Dortmund (IfADo), 44139 Dortmund, Germany; edlund@ifado.de (K.E.); Hengstler@ifado.de (J.G.H.); 4TRON-Translational Oncology at the University Medical Center Mainz, 55131 Mainz, Germany; sahin@uni-mainz.de; 5Bayer AG, 42113 Wuppertal, Germany; mathias.gehrmann@bayer.com

**Keywords:** interferon, breast cancer, prognosis, molecular subtypes

## Abstract

Interferons are crucial for adaptive immunity and play an important role in the immune landscape of breast cancer. Using microarray-based gene expression analysis, we examined the subtype-specific prognostic significance of interferon-γ (IFN-γ) as a single gene as well as an IFN-γ signature covering the signaling pathway in 461 breast cancer patients. Prognostic significance of IFN-γ, as well as the IFN-γ signature for metastasis-free survival (MFS), were examined using Kaplan–Meier as well as univariate and multivariate Cox regression analyses in the whole cohort and in different molecular subtypes. The independent prognostic significance of IFN-γ as a single gene was limited to basal-like breast cancer (hazard ratio (HR) 2.779, 95% confidence interval (95% CI) 1.117–6.919, *p* = 0.028). In contrast, the IFN-γ-associated gene signature was an independent prognostic factor in the whole cohort (HR 2.287, 95% CI 1.410–3.633, *p* < 0.001) as well as in the basal-like (HR 3.458, 95% CI 1.154–10.359, *p* = 0.027) and luminal B (HR 2.690, 95% CI 1.416–5.112, *p* = 0.003) molecular subtypes. These results underline the subtype-dependent prognostic influence of the immune system in early breast cancer.

## 1. Introduction

The immune system plays a decisive, but also ambivalent role in tumorigenesis as well as in tumor elimination of solid tumors [1]. Immune infiltrates, which are composed of different types of immune cells, are known to have a prognostic and predictive impact in breast cancer [2,3,4]. Dependent on the predominantly existing immune cell type and the microenvironment, this immune reaction can result in tumor rejection, but also in tumor progression [5]. The dynamic interaction between the immune system and the tumor leads to different stages of tumor evolution: elimination–equilibrium–escape [6]. In order to achieve the destruction of cancer cells by the immune system, different stages must be passed through step by step, which are collectively referred to as the cancer-immunity cycle [1]. In their function as cytokines, interferons have antiviral, antitumor, and immunomodulatory properties [7]. In particular, interferon-γ (IFN-γ) plays a crucial role in the regulation of antitumor immunity: mainly secreted by activated lymphocytes such as CD8 cytotoxic T-cells or CD4 T-helper cells type I (Th1), IFN-γ can enhance Th1-mediated antitumor immune response in terms of a positive feedback loop [7]. In contrast, IFN-γ is also known to play a protumorigenic role by transmitting antiapoptotic and proliferative signals, resulting in immune-escape of tumor cells. The downstream signaling pathway of IFN-γ is well-characterized [7]. Upon ligand binding, IFN-γ receptor 1 and 2 (IFNγR1 and IFNγR2) oligomerize and transphosphorylate activating Janus activated kinase (JAK) 1 and 2. Thereby, IFNγR1 is phosphorylated, creating a docking site for the signal transducer and activator of transcription (STAT) 1. The differential expression of IFNγR1 and IFNγR2 in the local microenvironment in turn controls the T-helper phenotype switch between Th1 and Th2, which may lead to a modulation of the subsequent immune response. 

Phosphorylated STAT1 homodimerizes and increases the transcription of primary response genes. An important primary response gene is the transcription factor interferon-regulatory factor 1 (IRF1), which acts as a transcription activator of interferon-stimulated response elements and leads to the transcription of a large number of secondary response genes (e.g., Fas-associated protein with death domain (FADD), tumor necrosis factor-related apoptosis-inducing ligand (TRAIL) caspase-8). Comprehensive genome-wide analyses identified apoptosis, DNA damage, and immune processes in breast cancer cells as the most enriched target processes underlying the direct tumoricidal property of this cytokine. In addition, IFN-γ controls gene expression programs that regulate a complex interaction of cytokine and chemokine receptors, cell activation markers, cellular adhesion proteins, the major histocompatibility complex (MHC), proteasome formation, protein turnover, and signal regulators. Overall, IFN-γ as a proinflammatory cytokine is involved in the maintenance of Th1 line binding and the inhibition of Th2 cell differentiation as well as in antiproliferative, antiangiogenic, and proapoptotic effects leading to antitumor effects [7]. 

Several studies demonstrated the prognostic and predictive impact of immune-related gene signatures, mainly focusing on T- or B-cells. For instance, Rody et al. showed a favorable prognostic effect of a T-cell metagene in estrogen receptor (ER)-negative as well as in human epidermal growth factor receptor 2 (HER2)-positive breast cancer [8]. In a previous gene expression study, we identified a B-cell metagene that was correlated with improved metastasis-free survival (MFS) in highly proliferating node-negative breast cancer patients regardless of ER or HER2 status [9]. Later, immunoglobulin kappa C (IGKC), as an important part of the B-cell metagene, was associated with both prognostic and predictive effects in early breast cancer [10]. Furthermore, in a retrospective analysis of the FinHER trial, the B-cell attracting chemokine leukocyte chemoattractant–ligand (C–X–C motif) 13 (CXCL13) was independently associated with prognosis in triple-negative breast cancer. Other studies have also confirmed that B-cell signatures had a prognostic impact, particularly in basal-like and HER2-positive breast cancer subtypes [11]. In a further analysis, we examined the prognostic impact of different immune signatures in node-negative breast cancer patients and demonstrated a prognostic effect of a B-cell and T-cell signature, especially in HER2-positive breast cancer [12].

The clustering of IFN-stimulated genes in breast cancer was described early on in unsupervised hierarchical analyses [13]. However, a possible prognostic or predictive effect of IFN-regulated genes was not initially investigated. Later, Weichselbaum and coworkers could show a predictive impact of an IFN-related gene signature with regard to the response to chemotherapy and the efficiency of radiation in breast cancer [14]. In addition, IFN-γ-related gene signatures predicted clinical response to programmed cell death protein 1 (PD-1)/programmed cell death 1 ligand 1 (PD-L1)-inhibiting therapeutic agents in breast cancer [15] and in other malignancies [16]. However, studies on the prognostic role of IFN-γ yielded contradictory results [17]. 

To obtain a complete picture of the prognostic role of IFN-γ in the immune landscape of breast cancer, we defined an IFN-γ signature that includes the abovementioned genes. This prompted us to evaluate the prognostic significance of IFN-γ and an IFN-γ-associated gene signature in a well-characterized cohort of 461 patients with early breast cancer. 

## 2. Results

### 2.1. Expression of IFN-γ and the IFN-γ Signature in Different Molecular Subtypes

In a cohort of 461 breast cancer patients, we evaluated the prognostic significance of IFN-γ as a single gene as well as an IFN-γ gene expression signature composed of IFN-γ and genes downstream in the IFN-γ pathway: IFN-γ receptor 1, IFN-γ receptor 2, the tyrosine kinases JAK1 and JAK2, as well as the transcription factors STAT1 and IRF1 [7] (Table 1). 

The boxplot diagrams in Figure 1 illustrate the distribution of gene expression of IFN-γ as a single gene (Figure 1a) and of the IFN-γ gene signature (Figure 1b) in the different molecular subtypes (i.e., luminal A, luminal B, HER2-positive, basal-like). The boxplot diagrams show that the expression of both IFN-γ and the IFN-γ signature is lowest in luminal A and highest in basal-like breast cancer (*p* < 0.001).

### 2.2. Prognostic Impact of IFN-γ as a Single Gene on MFS 

In the whole cohort, IFN-γ had no significant impact on MFS (*p* = 0.698, log rank; Figure 2). In univariate Cox regression analysis, IFN-γ as a single gene also failed to show a prognostic impact on the whole cohort (HR 1.084, 95% CI: 0.769–1.527, *p* = 0.646; Table 2); the prognostic impact of IFN-γ as a single gene was confined to the basal-like subtype (*p* = 0.033, log rank; Figure 3d). 

High tumor IFN-γ content was significantly associated with favorable MFS in basal-like breast cancer (HR 2.459, 95% CI 1.040–5.815, *p* = 0.040), unlike in the rest of the molecular subtypes. Tumor IFN-γ content also had an independent influence on MFS in a multivariate analysis adjusted for age, tumor size, axillary nodal status, and histological grade of differentiation (HR 2.779, 95% CI 1.117–6.919; *p* = 0.028; Table 3).

### 2.3. Impact of an IFN-γ Signature on MFS

Kaplan–Meier curves demonstrated the prognostic impact of the IFN-γ signature: higher expression of the IFN-γ signature was associated with a significantly longer MFS in the whole cohort (*p* = 0.012, log rank; Figure 4). 

This effect of the IFN-γ signature was confirmed in univariate Cox regression analysis (HR 1.554, 95% CI: 1.1099–2.199, *p* = 0.013; Table 2). In addition to the IFN-γ signature, univariate Cox regression analysis identified the following clinical–pathological factors as further prognostic markers: tumor size (HR 0.480, 95% CI 0.327–0.705, *p* < 0.001), lymph node status (HR 0.493, 95% CI: 0.344–0.706, *p* < 0.001), tumor grade (HR 0.217, 95% CI 0.089–0.531, *p* = 0.001), immunohistochemically determined estrogen receptor status (HR 1.954, 95% CI: 1.328–2.905, *p* = 0.001), progesterone receptor status (HR 1.843, 95% CI: 1.287–2.640, *p* = 0.001), HER2 status (HR 0.557, 95% CI: 0.336–0.926, *p* = 0.024), and the proliferation marker Ki-67 (HR 0.577, 95% CI: 0.389–0.886, *p* = 0.006; Table 2). 

In subtype analysis, the prognostic effect of the IFN-γ signature was particularly pronounced in the luminal B (*p* = 0.007, log rank), HER2-positive (*p* = 0.033, log rank), and basal-like molecular subtype (*p* = 0.050, log rank), but not in the Luminal A breast cancer samples (Figure 5). 

High expression of the IFN-γ signature had an independent influence on MFS in a multivariate analysis adjusted for age, tumor size, axillary nodal status, and histological grade of differentiation both in basal-like breast cancer (HR 3.458, 95% CI 1.154–10.359, *p* = 0.027) as well as luminal B tumors (HR 2.690, 95% CI 1.416–5.112, *p* = 0.003; Table 4).

In multivariate Cox regression analysis of the whole cohort of patients, the IFN-γ signature retained its prognostic impact (HR 2.287, 95% CI: 1.410–3.633, *p* < 0.001); furthermore, tumor size (HR 0.608, 95% CI: 0.378–0.979, *p* = 0.041), estrogen receptor status (HR 2.171, 95% CI:1.003–4.701, *p* = 0.049), and tumor grade (HR 0.310, 95% CI 0.110–0.870, *p* = 0.026) were further independent clinical pathological parameters (Table 5).

## 3. Discussion

We showed in this retrospective gene expression study that IFN-γ as a single gene had significant prognostic influence only in basal-like breast cancer. However, an IFN-γ signature covering the IFN-γ pathway had a prognostic impact on the entire cohort of 461 breast cancer patients with long-term follow-up: higher expression of the IFN-γ-signature was associated with better prognosis. Since IFN-γ is more implicated in antitumor activity than the type I interferons, we decided to focus on it. While IFN-γ is certainly an important member in the respective pathway, adding information from other pathway components improves the prognostic power. Apparently, a clinically relevant pathway is better captured by a multicomponent signature than by a single gene alone. Indeed, the IFN-γ signature retained its prognostic impact in multivariate analysis. The prognostic impact of the IFN-γ signature was particularly pronounced in the luminal B, HER2-positive, and basal-like subtypes, suggesting that the IFN-γ pathway plays an important role, particularly in highly proliferating tumors. This is in line with our results of previous gene expression studies evaluating the prognostic impact of different immune-related gene signatures. The B-cell metagene was correlated with improved outcomes in highly proliferating node-negative breast cancer patients regardless of ER or HER2 status [9]. In a further study evaluating CXCL13 mRNA expression in the FinHER trial, CXCL13 as a marker of the humoral immune system was associated with favorable prognosis, particularly in triple-negative breast cancer (TNBC) [11]. One possible explanation for the finding that immune-related gene signatures have prognostic and predictive effects, particularly in the case of strongly proliferating tumors, especially TNBC or basal-like breast cancer, is the occurrence of an increased mutation load and neoepitopes that can induce or enhance an antitumor immune response [18,19]. In fact, TNBC, which is not synonymous with basal-like breast cancer but largely overlaps with it, has a higher level of tumor-infiltrating lymphocytes [2,3]. By triggering protective antitumor immune reactions, e.g., via IFN-γ as effector cytokine, this might lead to an improved prognosis.

Interestingly, Callari et al. described, in a retrospective study using gene expression analyses, a significant impact of an interferon-induced metagene depending on ER and HER2 status. In ER-positive HER2-negative breast cancer, a significant correlation was found between a worse prognosis and higher expression of an IFN-correlated metagene, while in patients with HER2-positive breast cancer, a higher expression of the IFN metagene was associated with a better clinical outcome [20]. 

Our study has some strengths and limitations. A potential weakness is that our study was retrospective and performed at a single certified breast cancer center. Another limitation is that we have not been able to validate our microarray results with real-time polymerase chain reaction, as we report on patients treated more than 2 or 3 decades ago. Unfortunately, not enough tumor tissue is available for additional investigations. Nevertheless, we have previously reported a good correlation of both methods, which has led to the development of the prognostic breast cancer biomarker EndoPredict™ [21].

However, a major strength of our study is the consecutive inclusion of all patients with (i) an adequate amount of fresh-frozen tissue available for successful DNA microarray analysis with (ii) long-term follow-up and (iii) well-defined adjuvant treatment strategies. In this cohort, we show not only the prognostic significance of IFN-γ as a single gene in basal-like breast cancer but also an independent association of a signature covering the IFN-γ pathway in the entire cohort of 461 breast cancer patients with long-term follow-up. This finding underlines the prognostic role of the IFN-γ pathway in early breast cancer across different molecular subtypes. This favorable prognostic effect was particularly pronounced in rapidly proliferating molecular subtypes like luminal B and basal-like. In contrast, no prognostic effect was found for luminal A breast cancer.

In summary, we show, in a comprehensive overview of interferon-γ in the immune landscape of early breast cancer, the subtype-dependent prognostic role of interferon-γ and its signaling pathway. In particular, the results in the luminal B molecular subtype could pave the way for studies in which immune therapies like immune checkpoint inhibitors or personalized vaccination strategies can be used beyond estrogen-receptor-negative breast cancer.

## 4. Methods

### 4.1. Patient Characteristics and Tissue Specimens

Briefly, 461 patients with early breast cancer, who received surgery between 1986 to 2000 at the Department of Gynecology and Obstetrics of the University Medical Center Mainz, entered the study (Table 6). We included all consecutive patients with an adequate amount of fresh-frozen tumor tissue available for successful Affymetrix (Santa Clara, CA, USA) microarray analysis. The entire cohort consisted of three subgroups with different systemic treatments: 

(i) “N0 cohort”: 200 node-negative early breast cancer patients with no further adjuvant therapy after surgery and irradiation.

(ii) “tamoxifen cohort”: 165 patients treated with tamoxifen as sole adjuvant therapy.

(iii) “chemotherapy cohort”: 96 patients treated with either cyclophosphamide, methotrexate, fluorouracil (CMF; *n* = 34) or epirubicin, cyclophosphamide (EC; *n* = 62) without endocrine therapy in the adjuvant setting.

The established prognostic factors (histologic grade, tumor size, nodal status, age at diagnosis, ER, PR, HER2, and Ki-67) were collected from the pathology reports and the breast cancer database of our department. For all tumors, samples were snap-frozen and stored at −80°C. Tumor cell content exceeded 40% in all samples. Approximately 50 mg of frozen breast tumor tissue were crushed in liquid nitrogen. RLT buffer was added, and the homogenate was centrifuged through a QIAshredder column (Qiagen Hilden, Germany). From the eluate, total RNA was isolated with the RNeasy Kit (Qiagen, Hilden, Germany) according to the manufacturer’s instructions. RNA yield was determined by UV absorbance, and RNA quality was assessed by analysis of rRNA band integrity on an Agilent 2100 Bioanalyzer RNA 6000 LabChip kit (Agilent Technologies, Santa Clara, CA, USA), as previously described [9]. The study was approved by the ethical review board of the medical association of Rhineland-Palatinate.

The median age of the patients at diagnosis was 62 years. The mean follow-up time was 12 years; 132 patients (28.6%) developed distant metastases from breast cancer (Table 6).

### 4.2. Gene Expression Analysis

Fresh-frozen tumors (*n* = 461) obtained from the Department of Obstetrics and Gynecology of the University Medical Center Mainz were profiled on HG-U133A arrays (Affymetrix, Santa Clara, CA, USA) to quantify the relative transcript abundance in the breast cancer tissue, as previously described [9I. Briefly, starting from 5 μg total RNA, labeled cRNA was prepared using the Roche Microarray cDNA Synthesis, Microarray RNA Target Synthesis (T7), and Microarray Target Purification kits (Roche Applied Science, Mannheim, Germany), according to the manufacturer’s instructions. Raw expression data (CEL files) were normalized using frozen robust multiarray analysis (fRMA). In the analysis settings, the global scaling procedure was chosen, which multiplied the output signal intensities of each array to a mean target intensity of 500. Samples with suboptimal average signal intensities (i.e., scaling factors >25) or glyceraldehyde-3-phosphate dehydrogenase 3′/5′ ratios >5 were relabeled and rehybridized on new arrays. Throughout the text below, all expression values and respective thresholds from fresh frozen material measured by HG-U133A arrays refer to TGT500 scaling. The majority of the samples had already been deposited at the National Center for Biotechnology Information (NCBI) Gene Expression Omnibus (GEO) under the accession numbers GSE11121 and GSE26971. Since these datasets were submitted in 2008 and 2011, we decided to file the complete record of 461 samples used in the current study with updated follow-up at the NCBI in the GEO database under accession number GSE158309.

### 4.3. IFN-γ Signature

The IFN-γ signaling pathway signature consists of the following genes with corresponding probesets: interferon-γ (IFNG (210354_at)), interferon-γ receptor 1 (IFNGR1 (202727_s_at, 211676_s_at)), interferon-γ receptor 2 (IFNGR2 (201642_at)), janus kinase 1 (JAK1 (201648_at)), janus kinase 2 (JAK2 (205841_at, 205842_s_at)), signal transducer and activator of transcription 1 (STAT1 (200887_s_at, 209969_s_at)), and interferon regulatory factor 1 (IRF1 (202531_at)). It was calculated as representative of all genes contained within this signature based on the median of the normalized expression values. To dichotomize, values above the median of the IFN-γ signature were defined as high expression, whereas values below the median as low expression.

### 4.4. Molecular Subtypes

The determination of intrinsic subtypes was performed according to Haibe-Kains and coworkers [22], who postulated a three-gene model with the estrogen receptor gene (ESR1), HER2, and aurora kinase A (AURKA). Briefly, ER and HER2 status were derived from the bimodally distributed mRNA levels of the corresponding genes (probesets: ESR1 205225_at and ERBB2 216836_s_at) based on fRMA normalized expression values. The cut-off for ESR1 was determined by model-based clustering. The cut-off for ERBB2 was selected by the upper quartile plus interquartile range of the mRNA level. For AURKA, the median of the mRNA expression of the corresponding probe set (208079_s_at) was used as a cut-off, as previously described [12].

This procedure resulted in the following molecular subtypes:-ESR1-positive, HER2 negative, low proliferation (AURKA low) → luminal A-like-ESR1-positive, HER2 negative, high proliferation (AURKA high) → luminal B-like-HER2-positive-ESR1 negative, HER2-negative → basal-like

Table 6 shows the absolute and relative frequencies of the molecular subtypes (which were determined based on gene expression data) within the investigated cohort of patients.

### 4.5. Statistical Analysis

Statistical analyses were performed using the SPSS statistical software program, version 23.0 (SPSS Inc, Chicago, IL, USA) and R. Patients’ characteristics were given in absolute and relative numbers (Table 3). Differences in the expression of IFN-γ and the IFN-γ signature between molecular subtypes were assessed using analysis of variance (ANOVA) and posthoc Tukey’s test. The prognostic significance of IFN-γ as a single gene and an IFN-γ signature for MFS was examined by Kaplan–Meier survival analysis (≤median vs. >median) as well as univariate and multivariate Cox regression analysis. This was adjusted for pT stage (T1, -2 vs. T3, -4), histological grade (GI + GII vs. GIII), and immunohistochemically determined ER (negative vs. positive), PR (negative vs. positive), HER2 (negative vs. positive), and Ki-67 (≤20% vs. >20%). The significance of Kaplan–Meier survival analysis was assessed by the *p*-value of the log-rank test. All *p*-values are two-sided. As no correction for multiple testing was carried out, these are descriptive measures.

## Figures and Tables

**Figure 1 ijms-21-07178-f001:**
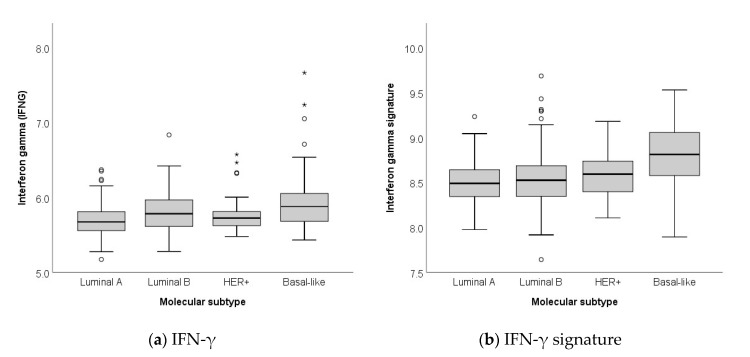
Distribution of gene expression data of (**a**) IFN-γ and (**b**) IFN-γ gene signature, dependent on molecular subtypes. ⭑, ⚬ indicate outliers in a boxplot.

**Figure 2 ijms-21-07178-f002:**
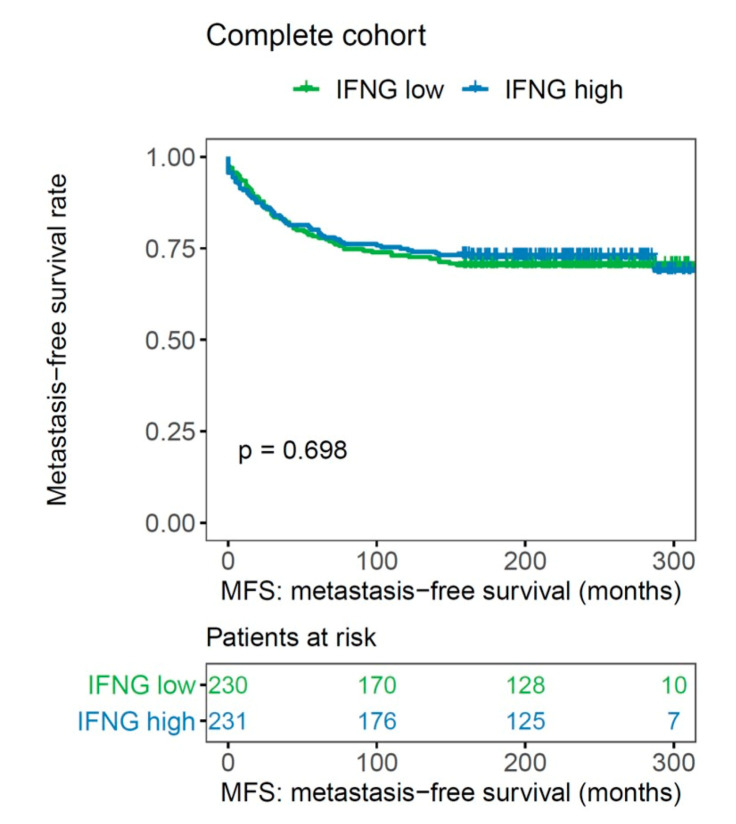
Association of IFN-γ with metastasis-free survival in the whole cohort of early breast cancer patients (*n* = 461).

**Figure 3 ijms-21-07178-f003:**
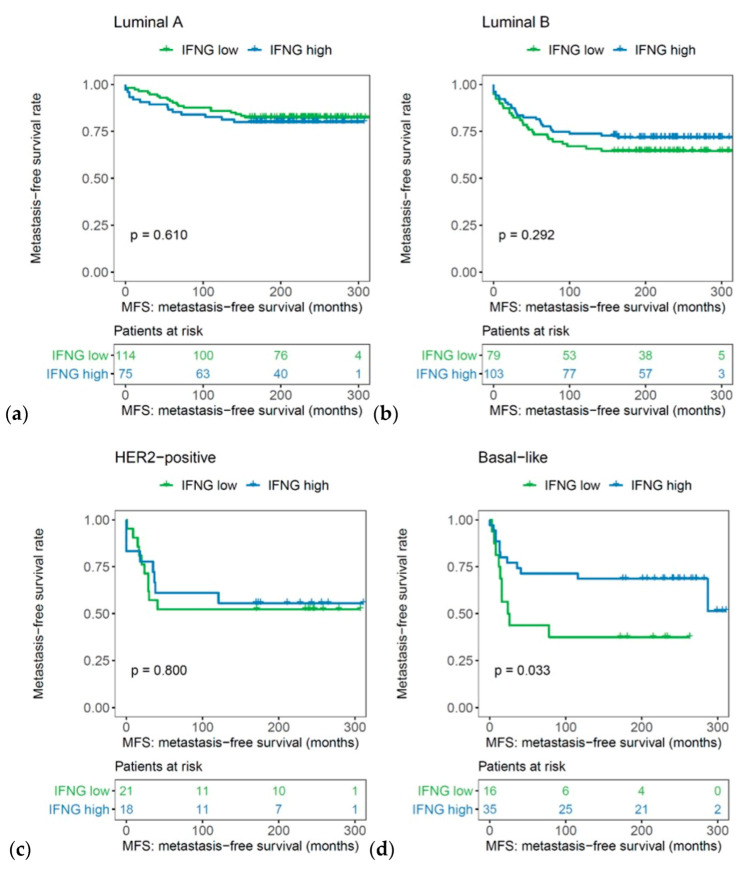
Association of IFN-γ, with metastasis-free survival in (**a**) luminal A, (**b**) luminal B, (**c**) HER2-positive, and (**d**) basal-like molecular subtypes (*n* = 461).

**Figure 4 ijms-21-07178-f004:**
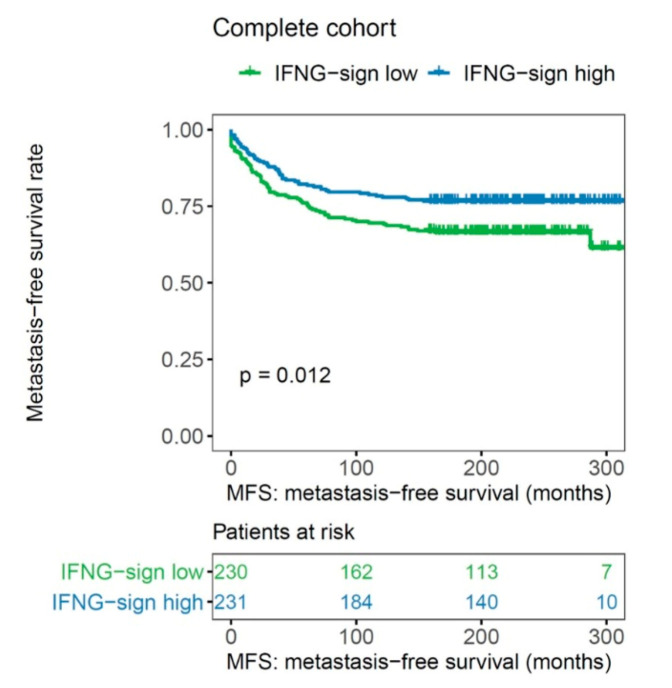
Association of the IFN-γ signature with metastasis-free survival in the whole cohort of early breast cancer patients (*n* = 461).

**Figure 5 ijms-21-07178-f005:**
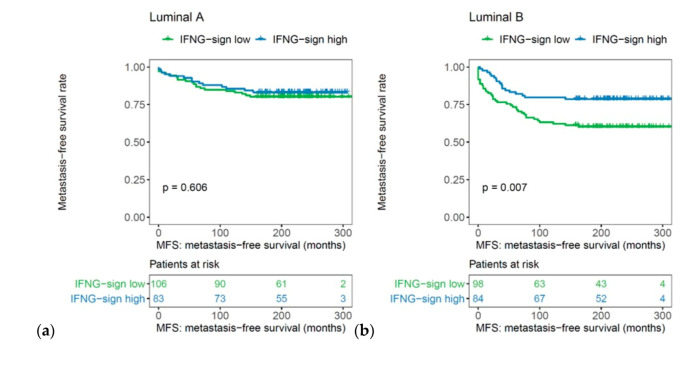
Association of IFN-γ signature with metastasis-free survival in (**a**) luminal A, (**b**) luminal B, (**c**) HER2-positive, and (**d**) basal-like molecular subtypes (*n* = 461).

**Table 1 ijms-21-07178-t001:** Genes and probesets belonging to the IFN-γ signature.

Gene	Gene Symbol	Probeset
Interferon-γ	IFNG	210354_at
Interferon-γ receptor 1	IFNGR1	202727_s_at 211676_s_at
Interferon-γ receptor 2	IFNGR2	201642_at
Interferon regulatory factor 1	IRF1	202531_at
Janus kinase 1	JAK1	201648_at
Janus kinase 2	JAK2	205841_at 205842_s_at
Signal transducer and activator of transcription 1	STAT1	200887_s_at 209969_s_at

**Table 2 ijms-21-07178-t002:** Univariate cox regression analysis for metastasis-free survival (*n* = 461).

		HR	95% CI		*p*-Value
			Lower	Upper	
IFN-γ-signature	Low vs. high	1.554	1.099	2.199	0.013
IFN-γ	Low vs. High expression	1.084	0.769	1.527	0.646
Age	</=50 vs. >50	1.297	0.881	1.909	0.188
T	T1 vs. T2, T3,4	0.480	0.327	0.705	<0.001
N	N0 vs. N1,2,3	0.493	0.344	0.706	<0.001
Grade	GI/II vs. III	0.217	0.089	0.531	0.001
ER	Neg. vs. pos.	1.954	1.328	2.905	0.001
PR	Neg. vs. pos.	1.843	1.287	2.640	0.001
HER2	Neg. vs. pos.	0.557	0.336	0.926	0.024
Ki-67	<20% vs. >20%	0.577	0.389	0.886	0.006

Abbreviations: 95% CI, 95% confidence interval; ER, estrogen receptor; HER2, human epidermal growth factor receptor; HR, hazard ratio; IFN-γ, interferon-γ; N, nodal-status; PR, progesterone receptor; T, tumor size.

**Table 3 ijms-21-07178-t003:** Association between IFN-γ and MFS in molecular subtypes using univariate and multivariate Cox regression analysis adjusted for age at diagnosis, tumor size, nodal status, and grading.

Subtype	Univariate Model		Multivariate Model	
	HR (95% CI)	*p*	HR (95% CI)	*p*
Luminal A-like	0.841 (0.430–1.642)	0.611	1.066 (0.477–2.380)	0.877
Luminal B-like	1.320 (0.785–2.218)	0.295	1.659 (0.909–3.028)	0.099
HER2-positive	1.127 (0.444–2.861)	0.801	1.265 (0.460–3.475)	0.649
Basal-like	2.459 (1.040–5.815)	0.040	2.779 (1.117–6.919)	0.028

Abbreviations: CI, confidence interval; HR, hazard ratio.

**Table 4 ijms-21-07178-t004:** Association between IFN-γ signature and MFS in molecular subtypes using univariate and multivariate Cox regression analysis adjusted for age at diagnosis, tumor size, nodal status, and grading.

Subtype	Univariate Model		Multivariate Model	
	HR (95% CI)	*p*	HR (95% CI)	*p*
Luminal A-like	1.194 (0.607–2.348)	0.607	1.314 (0.596–2.898)	0.498
Luminal B-like	2.109 (1.206–3.688)	0.009	2.690 (1.416–5.112)	0.003
HER2-positive	2.669 (1.042–6.840)	0.041	1.925 (0.620–5.978)	0.257
Basal-like	2.355 (0.972–5.707)	0.058	3.458 (1.154–10.359)	0.027

Abbreviations: CI, confidence interval; HR, hazard ratio.

**Table 5 ijms-21-07178-t005:** Multivariate cox regression analysis for metastasis-free survival (*n* = 461).

		HR	95% CI		*p*-Value
			Lower	Upper	
IFN-γ signature	Low vs. high	2.287	1.440	3.633	<0.001
Ki-67	<20% vs. >20%	0.680	0.431	1.073	0.098
T	T1 vs. T2,3,4	0.608	0.378	0.979	0.041
N	N0 vs. N1,2,3	0.982	0.621	1.554	0.939
Grade	GI/II vs. III	0.310	0.110	0.870	0.026
ER	Neg. vs. pos.	2.171	1.003	4.701	0.049
PR	Neg. vs. pos.	0.997	0.506	1.965	0.993
HER2	Neg. vs. pos.	0.329	0.410	1.348	0.329

Abbreviations: 95% CI, 95% confidence interval; ER, estrogen receptor; HER2, human epidermal growth factor receptor; HR, hazard ratio; IFN-γ, interferon-γ; N, nodal-status; PR, progesterone receptor; T, tumor size.

**Table 6 ijms-21-07178-t006:** Patients characteristics (*n* = 461).

	Number of Patients (*n* = 461)	Percentage (%)
Age at diagnosis		
≤50	104	22.6
>50	357	77.4
Tumor size		
T1	188	40.8
T2	214	46.4
T3	19	4.1
T4	39	8.5
missing value	1	0.2
Tumor grade		
GI	62	13.4
GII	261	56.6
GIII	106	23
missing value	32	6.9
Lymph node status		
N0	253	54.9
N1	138	29.9
N2	49	10.6
missing value	21	4.6
Tumor type		
Invasive ductal (NST)	291	63.1
Invasive lobular	79	17.1
others	91	19.7
ER		
positive	381	82.6
negative	79	17.1
missing value	1	0.2
PR		
positive	346	75.1
negative	114	24.7
missing value	1	0.2
HER2		
positive	46	10
negative	358	77.7
missing value	57	12.3
Ki-67		
>20%	138	29.9
≤20%	250	54.2
missing value	73	15.8
Molecular subtypes		
Luminal A	189	41
Luminal B	182	39.5
Basal-like	51	11.1
HER2-positive	39	8.5
Distant metastasis		
Yes	132	28.6
No	329	71.4
Treatment cohort		
N0, untreated	200	43.4
tamoxifen	165	35.8
chemotherapy	96	20.8
CMF	34	7.4
EC	62	13.4

Abbreviations: CMF, cyclophosphamide/methotrexate/5-fluorouracil; EC, epirubicin/cyclophosphamide; ER, estrogen receptor; HER2, human epidermal growth factor receptor; IFN-γ, interferon-γ; N, nodal-status; NST, no special type; PR, progesterone receptor; T, tumor size.

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
