# Peer review of "Prognostic Significance of Interferon-γ and Its Signaling Pathway in Early Breast Cancer Depends on the Molecular Subtypes"

_ijms, 2020, doi:10.3390/ijms21197178_

Round 1

Reviewer 1 Report

The paper Prognostic significance of interferon-g and its signaling pathway in early breast cancer depends on the molecular subtypes offers an interesting insight into the prognosis of early breast cancer using IFN-gamma related genes as a tool to stratify the metastasis-free survival of oncologic patients. The paper is very well written, and it address a very interesting point, however there are some missing experiments that have to be performed before publication.

MAJOR REVISION:

  1. It could be useful indicate in the first paragraph of Results, a table in which the authors should show the list of the main genes belonging to IFN-g signature
  2. The authors should confirm, by Real-Time PCR some of these differential expressed genes, in order to confirm the microarray analysis.
  3. The authors should discuss also the reasons why the IFN-g signature seems to be higher in the basal-like tumors and at the same time it could be related to better prognosis.

MINOR REVISION:

    1. IFN-gamma pathway has intimate connections with other important pathways (such as IFN-alpha and IFN-beta). Plus, STAT1, once activated, can also recruit STAT3 to transcribe IFN-related genes. The authors should motivate why they decided to exclude these genes from their analysis in the discussion.
  • In the introduction, the authors mention the primary and the secondary response genes activated by IFN-gamma. The authors should expand this part, presenting the main genes activated and their role in anti-tumoral immunity.
  1. All the title of the Results section should be in bold characters
  2. It could be more simple to understand the figures if the authors put together all the graphs belonging to the same number figure (e.g: the figure 2 and figure 5 should be in the same page)

Author Response

Dear Sir or Lady,

We thank reviewer 1 for his thorough revision and appreciate the justified comments that have improved our manuscript. 

We hope that the revised manuscript is acceptable for publication in the International Jpurnal of Molecular Sciences.

With kind regards,

Marcus Schmidt

Reviewer 2 Report

In this paper “Prognostic significance of interferon-γ and its signaling pathway in early breast cancer depends on the molecular subtypes.” Anne-Sophie Heimes and coworkers performed an analysis of subtype specific prognostic significance of IFN-γ as a single gene as well as an IFN- γ -signature covering the signalling pathway in 461 breast cancer patients. These examination was conducted by using microarray-based gene expression analysis. The authors observed the subtype-dependent prognostic influence of the immune system in early breast cancer. Performed analysis are very interesting however, there are some issues that need to be addressed before a manuscript can be recommended for publication.

1). First at all, as Authors mentioned the entire cohort consisted of three subgroups with different systemic treatment. It is well known that anti-cancer therapy can modulate the immune system response. Taking into account the above fact, and the fact that the individual subgroups are quite numerous, I suggest to additionally analyze the data divided into two subgroups: with no adjuvent thepary (N0 cohort) and with adjuvent therapy (tamoxifen cohort + chemotherapy cohort)

2). Next, figures should be refined. E.g. the numbers describing the axes are barely visible, the survival analysis figures may be smaller and should be of better quality; Axis signatures are in years (100, 200, 300).

3). Why the authors performed an IFN- γ as a single gene on MFS analysis with subgroups (lumianl A and B, HER2-positive and basal-like) based on only one test, i.e. Kaplan-Meier analysis. I couldn't find Cox regression analysis.

4). In lines 138-140, which statistical test was used?

5). Table 3 should be redeveloped. E.g. which means </ =; what the dots in the "treatment collective" section mean; the numers in the "number of patients" column for the "tumor grade", "molecular subtypes" and "treatment collective" sections, are shifted.

Author Response

Dear Sir or Lady,

we thank reviewer 2 for the thorough revision and the justified comments that have improved our manuscript.

We hope that the revised manuscript is acceptable for publication in the International Journal of Molecular Sciences.

With kind regards,

Marcus Schmidt

Round 2

Reviewer 1 Report

During this revision, the authors clarify several points that I suggested in the previous form.

However, there are very low minor points that should be fix.

1) row 78: γ shoukd be insert after IFN

2) row 80: the sentence within () should be deleted and only the references should be maintain--

3) rows 81-82: this sentence should be moved in the last part of introduction after the sentence present in the row 104-

4) The authors should explain also, in the Introduction, the role of some other genes that are listed in the table 1

5) Rows: 278-281: I should correct in this way: "In fact, TNBC which is not synonymous with basal-like breast cancer but largely overlaps with it, has a higher level of tumor-infiltrating lymphocytes [2,3]. By triggering protective anti-tumor immune reactions, e.g. via IFN-g as effector cytokine, this might lead to an improved prognosis.“

6) Rows 261-263: this sentence should be clarify; I suggest "Since IFN-g is more implicated in the antitumor activity than the type I interferons, we decided to focus on it“.

7) When the authors discuss the limitations of their works should be discuss also the reason why they could not confirm their data with Real-Time PCR.

Author Response

Reviewer #1:

During this revision, the authors clarify several points that I suggested in the previous form.

However, there are very low minor points that should be fix.

1) row 78: γ shoukd be insert after IFN

Response: We inserted g after IFN:

“Overall, IFN-g as a proinflammatory cytokine…”

2) row 80: the sentence within () should be deleted and only the references should be maintain—

Response: We deleted this sentence.

“…leading to anti-tumor effects [7].”

3) rows 81-82: this sentence should be moved in the last part of introduction after the sentence present in the row 104-

Response: We moved the sentence in the last part oft he introduction:

“To obtain a complete picture of the prognostic role of IFN-g in the immune landscape of breast cancer we defined an IFN-g-signature that includes the above-mentioned genes. This prompted us to evaluate the prognostic significance of IFN-g and an IFN-g associated gene signature in a well-characterized cohort of 461 patients with early breast cancer. “

4) The authors should explain also, in the Introduction, the role of some other genes that are listed in the table 1

Response: The reviewer is right with the suggestion to decribe the role of some other genes listed in table 1. The role of several genes (e.g. IFN-g, JAK1, JAK2, STAT1, IRE) in the interferon-g signaling pathway has already been briefly mentioned in the introduction. We added the following sentence in the paragraph briefly outlining the role of signaling pathway in the introduction:

„The differential expression of IFNγR1 and IFNγR2 in the local microenvironment in turn controls the T-helper phenotype switch between Th1 and Th2, which may lead to a modulation of the subsequent immune response.“

5) Rows: 278-281: I should correct in this way: "In fact, TNBC which is not synonymous with basal-like breast cancer but largely overlaps with it, has a higher level of tumor-infiltrating lymphocytes [2,3]. By triggering protective anti-tumor immune reactions, e.g. via IFN-g as effector cytokine, this might lead to an improved prognosis.“

Response: Agreeing with the reviewer, we corrected this sentence:

"In fact, TNBC which is not synonymous with basal-like breast cancer but largely overlaps with it, has a higher level of tumor-infiltrating lymphocytes [2,3]. By triggering protective anti-tumor immune reactions, e.g. via IFN-g as effector cytokine, this might lead to an improved prognosis.

6) Rows 261-263: this sentence should be clarify; I suggest "Since IFN-g is more implicated in the antitumor activity than the type I interferons, we decided to focus on it“.

Response: We welcome this suggestion and clarified this sentence accordingly:

“Since IFN-g is more implicated in the antitumor activity than the type I interferons, we decided to focus on it.“

7) When the authors discuss the limitations of their works should be discuss also the reason why they could not confirm their data with Real-Time PCR.

Response: The reviewer is right. We included the following sentences and an additional reference:

„Another limitation is that we have not been able to validate our microarray results with real-time polymerase chain reaction, as we report on patients treated more than 2 or 3 decades ago. Unfortunately, not enough tumor tissue is available for additional investigations. Nevertheless, we have previously reported a good correlation of both methods, which has led to the development of the prognostic breast cancer biomarker EndoPredict™ [22].“

Reviewer 2 Report

I appreciate the authors put much effort to fix the issues that were mentioned in the original review. The corrections implemented by the Authors improve the quality of the manuscript.

Author Response

Reviewer 2

I appreciate the authors put much effort to fix the issues that were mentioned in the original review. The corrections implemented by the Authors improve the quality of the manuscript.

Response: We very much appreciate the reviewers' assessment.